# Sentinel Node Biopsy after Neoadjuvant Chemotherapy for Breast Cancer: Preliminary Experience with Clinically Node Negative Patients after Systemic Treatment

**DOI:** 10.3390/jpm11030172

**Published:** 2021-03-02

**Authors:** Alejandro Martin Sanchez, Daniela Terribile, Antonio Franco, Annamaria Martullo, Armando Orlandi, Stefano Magno, Alba Di Leone, Francesca Moschella, Maria Natale, Sabatino D’Archi, Lorenzo Scardina, Elena J. Mason, Flavia De Lauretis, Fabio Marazzi, Riccardo Masetti, Gianluca Franceschini

**Affiliations:** 1Multidisciplinary Breast Center, Dipartimento Scienze della Salute della donna e del Bambino e di Sanità Pubblica, Fondazione Policlinico Universitario A. Gemelli IRCCS, Largo Agostino Gemelli 8, 00168 Rome, Italy; daniela.terribile@policlinicogemelli.it (D.T.); antonio.franco89@icloud.com (A.F.); annamaria_martullo@virgilio.it (A.M.); armando.orlandi@policlinicogemelli.it (A.O.); stefano.magno@policlinicogemelli.it (S.M.); albadileone@policliunicogemelli.it (A.D.L.); francesca.moschella@policlinicogemelli.it (F.M.); maria.natale@policlinicogemelli.it (M.N.); sabatinodarchi@gmail.com (S.D.); lorenzoscardina@libero.it (L.S.); elenajanemason@gmail.com (E.J.M.); flavia.delauretis@gmail.com (F.D.L.); riccardomasetti@policlinicogemelli.it (R.M.); gianluca.franceschini@policlinicogemelli.it (G.F.); 2Istituto di Semeiotica Chirurgica, Università Cattolica del Sacro Cuore, 1, 00168 Rome, Italy; 3Division of Medical Oncology, Fondazione Policlinico Universitario A. Gemelli IRCCS, 00168 Rome, Italy; 4Division of Radiotherapy, Dipartimento di Diagnostica per Immagini, Radioterapia Oncologica ed Ematologia, Fondazione Policlinico Universitario A. Gemelli IRCCS, 00168 Rome, Italy; fabio.marazzi@policlinicogemelli.it

**Keywords:** neoadjuvant chemotherapy, sentinel lymph node, breast cancer, systemic treatment, locally advanced breast cancer, mini-invasive treatment

## Abstract

Sentinel lymph node biopsy (SLNB) following neoadjuvant treatment (NACT) has been questioned by many studies that reported heterogeneous identification (IR) and false negative rates (FNR). As a result, some patients receive axillary lymph node dissection (ALND) regardless of response to NACT, leading to a potential overtreatment. To better assess reliability and clinical significance of SLNB status on ycN0 patients, we retrospectively analyzed oncological outcomes of 399 patients treated between January 2016 and December 2019 that were either cN0-ycN0 (219 patients) or cN1/2-ycN0 (180 patients). The Endpoints of our study were to assess, furthermore than IR: oncological outcomes as Overall Survival (OS); Distant Disease Free Survival (DDFS); and Regional Disease Free Survival (RDFS) according to SLNB status. SLN identification rate was 96.8% (98.2% in patients cN0-ycN0 and 95.2% in patients cN+-ycN0). A median number of three lymph nodes were identified and removed. Among cN0-ycN0 patients, 149 (68%) were confirmed ypN0(sn), whereas regarding cN1/2-ycN0 cases 86 (47.8%) confirmed an effective downstaging to ypN0. Three year OS, DDFS and RDFS were significantly related to SLNB positivity. Our data seemed to confirm SLNB feasibility following NACT in ycN0 patients, furthermore reinforcing its predictive role in a short observation timing.

## 1. Introduction

Sentinel lymph node biopsy (SLNB) is considered the gold standard for axillary staging in early breast cancer patients with clinically negative lymph nodes (cN0), as it reduces potential complications of axillary dissection (ALND) [1,2,3,4].

Since many studies have shown a great variation in identification (IR) and false negative rates (FNR), the reliability of SLNB after neoadjuvant chemotherapy (NACT) remains questionable [5,6,7,8,9,10].

As a result, several patients continue to undergo complete axillary dissection, regardless of axillary staging and response to neoadjuvant chemotherapy, leading to a potential overtreatment of both cN0 and cN1/2 patients who remained or became ycN0 after NACT [3].

For this particular subgroup of patients, recent studies reported acceptable IR and FNR, suggesting that SLNB could be feasible in cN0-ycN0 patients and also in women who are cN1/2 before chemotherapy and achieve an ycN0 status [10,11,12].

Moreover, for cN0-ycN0 patients, a recently published retrospective study correlates a metastatic SLN with a significant worsening of oncological outcomes, such as Distant Disease Free Survival (DDFS), proving that SLNB is not only feasible after NACT, but that in this setting it could be a good predictive tool to better assess patients at risk [13].

The aim of this analysis, besides reporting our personal workout model for patients receiving neoadjuvant regimens, is to better assess the feasibility and prognostic significance (according to status) of SLNB in ycN0 patients.

With this purpose, we retrospectively analyzed clinical and oncological results obtained from cT1-4 breast cancer patients who were either cN0 or cN1/2 prior to neoadjuvant treatment and became or remained cN0 at the end of the systemic therapy (ycN0).

The endpoints of our study were to evaluate, furthermore than IR: oncological outcomes as Overall Survival (OS); Distant Disease Free Survival (DDFS); and Regional Disease Free Survival (RDFS) according to SLNB status.

## 2. Materials and Methods

From the prospectively maintained database of the Multidisciplinary Breast Center of the Fondazione Policlinico Universitario Agostino Gemelli IRCCS in Rome, we identified patients with locally advanced breast cancer (cT1-cT4 patients, cN0-cN1/2) who had received neoadjuvant chemotherapy and remained or became ycN0, subsequently undergoing breast surgery and SLNB, between 2016 and 2019. 

We excluded from our analysis ycN0 patients in whom a SLN was not identified during surgical procedure, who consequently underwent direct ALND. 

Endpoints of our study were:“Overall Survival”: time from day of surgery to death from any cause or latest follow-up.“Distant Disease Free Survival”: time from day of surgery to distant recurrence.“Regional Disease Free Survival”: time from day of surgery to ipsilateral breast and/or axillary recurrence.

## 3. Clinical Workout 

The indication for neoadjuvant treatment (chemotherapy or endocrine therapy) and surgical management of the axilla were discussed during a multidisciplinary meeting (MDM) of breast surgeons, medical oncologists, radiation oncologists, radiologists, pathologists and geneticists. 

According to national and international guidelines Associazione Italiana di Oncologia Medica (AIOM) 2019 and National Comprehensive Cancer Network (NCCN) 2020), patients underwent NACT in the following cases:Patients with locally advanced breast cancer;Patients with operable breast cancer and an unfavorable breast volume/tumor size ratio, in order to reduce the tumor diameter and achieve a conservative treatment instead of mastectomy;Patients with operable breast cancer and clinically involved lymph nodes (cN+), with the aim of ensuring a SLNB instead of a direct ALND;Young patients with unfavorable risk factors (triple negative tumor, Human Epidermal growth factor—2: HER2+, high Ki-67 rates), to provide prompt systemic treatment.

Pre-neoadjuvant clinical staging: Locoregional staging was assessed by clinical examination, breast and axillary ultrasound, mammography, breast magnetic resonance, or core biopsy of both breast lesion and suspected axillary lymph nodes. 

The systemic staging was assessed by total body computed tomography scan or positron emission tomography and bone scintigraphy.

Neoadjuvant regimens: NACT regimen depended on stage and tumor characteristics.

We used the following chemotherapy schemes:HER2 negative patients:
-Sequential scheme: Anthracyclines plus Cyclophosphamide on day 1 every 21 days for 4 cycles (4 AC); followed by docetaxel on day 1 every 21 days for 4 cycles or paclitaxel on day 1 every week for 12 cycles.-6 TAC: docetaxel plus Doxorubicin plus Cyclophosphamide on day 1 every 21 days for 6 cycles.HER2 positive patients:
-6 TCH: docetaxel plus Carboplatin plus Herceptin on day 1 every 21 days for 6 cycles.-Sequential scheme: Anthracyclines plus Cyclophosphamide on day 1 every 21 days for 4 cycles (4 AC); followed by docetaxel on day 1 every 21 days for 4 cycles or paclitaxel on day 1 every week for 12 cycles plus Herceptin on day 1 every 21 days for 18 cycles.

Hormone therapy with aromatase inhibitor was delivered to elder and fragile postmenopausal patients with locally advanced breast cancer expressing hormone receptors (ER, PgR) and low Ki-67 (Luminal A and Luminal B). Neoadjuvant protocol was administered for at least six months. 

Clinical assessments during and after NACT: Before each cycle of chemotherapy, patients underwent treatment response monitoring with a clinical examination and “in office” breast/axillary ultrasound. 

Patients with no evidence of clinical response or with disease progression were the subject of multidisciplinary discussion about a change in NACT scheme or immediate surgery.

One month after NACT finalization, loco-regional staging was repeated (clinical examination, breast and axillary ultrasound, mammography, breast magnetic resonance).

Breast surgical treatment: Surgical management was discussed during a dedicated MDM, taking into account the clinical restaging and patient’s preferences. 

Patients with a favorable ratio between breast volume and residual lesion were addressed by conservative techniques:Level I oncoplastic breast surgery techniques—for resection of <20% of breast volume (peri-areolar, axillary or inframammary fold incisions).Level II oncoplastic surgery which involves resection of >20% of breast volume (round block, batwing and reduction mammoplasty techniques) [14].

In case of unfavorable ratio between breast volume and residual tumor size, multicentric cancer, inflammatory cancer and contraindications to adjuvant radiotherapy patients were judged eligible for mastectomy techniques and immediate breast reconstruction (implant or autologous reconstruction):“Nipple Sparing Mastectomy” (NSM—removal of all the breast glandular tissue, while the nipple and areola are left in place along with breast skin) if tumor did not involve the nipple or tissue under the areola.“Skin Sparing Mastectomy” (removal of breast glandular tissue, nipple and areola while breast skin is kept intact) if tumor involved the nipple–areola complex.Simple mastectomy (removal of breast glandular tissue, nipple, areola and breast skin) if tumor involved breast skin.

Axillary assessment: Axillary workout is summarized in Figure 1. 

SLNB was performed using blue dye technique (Patent Blue V or Methylene blue, 2–5 cc) injected sub-dermally, 15–30 min before surgery. Blue-stained axillary lymph nodes were defined as SLNs. Axillary lymph nodes whose consistency and dimension were considered suspicious were also removed and analyzed.

Pathologic examination of the SLN was macroscopic, cytologic and histologic. 

The intraoperative cytology examination of the lymph nodes was performed by dissecting them in two parts along the major axis of the capsule if larger than 0.5 cm. After SLN division, a slide was affixed or dragged on the cut surface of both halves and stained with Harris hematoxylin solution. 

In case of suspected cytology, lymph node halves were frozen to −22 °C, serially divided in ultrathin sections and stained with Harris hematoxylin solution. 

For definitive pathologic assessment, SLN was included and examined along with two consecutive sections stained with Hematoxilyn and Eosin (HE) and, subsequently, with five sequences of three consecutive sections, 200 microns spaced. The middle section of each series was colored with CAM5.2, and those remaining with HE. 

All non-sentinel nodes were examined with standard procedure, as mentioned for SLN intraoperative histologic assessment.

For patients with ypN+ (micro or micro-metastatic) disease, axillary dissection was directly performed. I and II level lymph nodes were always removed, while III level lymph nodes were removed only in case of intraoperative detection of clinically suspicious nodes at lower levels. 

Patients with isolated tumor cells positivity at SLNB were treated as ypN0 and did not receive ALND. 

Adjuvant treatments: were determined on the basis of patient’s age, pre-neoadjuvant clinical staging, surgical intervention, pathological staging and tumoral biology.

Adjuvant chemotherapy: Patients who did not make a pathological complete response to neoadjuvant treatment were treated according to different adjuvant regimens. 

Anthracyclines and/or Taxanes were given to patients who did not receive them in the neoadjuvant regimen.Triple negative patients were given Capecitabine;HER2 positive cancers were treated with Trastuzumab emtansine (TDM-1).Cancers expressing hormone receptors (estrogen receptor, progesterone receptor) were treated with selective estrogen receptor modulators (Tamoxifen) or Luteinizing Hormone Release Hormone analogues (Enantone, Decapeptyl) if in premenopausal age while postmenopausal patients were given aromatase inhibitors (Anastrozole, Letrozole, Exemestane).

Adjuvant radiotherapy: was tailored to the type of surgical intervention and pathological staging. Radiation was delivered using 3D conformal schemes and intensity modulated radiotherapy on linear accelerator using 6-10-15 MV photons.

Axillary radiation was considered for patients with pathologically positive lymph nodes and subsequent ALND with less than 10 nodes removed, ypN3 tumor staging, extracapsular invasion or isolated tumor cells (ITC) in SLNs.

## 4. Statistical Analysis

Statistical analysis was performed with SPSS version 26.0 for Windows. Results are expressed as mean, median and range. Fisher exact test was used for categorical variables. A *p* < 0.05 was considered statistically significant. 

Kaplan-Meier curves were used to plot OS, DDFS and RDFS. Oncological outcomes were calculated over a median follow up of 24 months (2–48).

Only factors significantly associated with this outcome in the univariate analyses were included in multivariate models. Multivariate analyses were performed on all patients, and separately for cases cN0 and cN1/2 prior to neoadjuvant treatment.

## 5. Results

Between January 2016 and December 2019, 4478 patients with invasive breast cancer were treated in our multi-disciplinary center. 

From our prospectively maintained database we extracted 412 patients with cT1-cT4 and cN0-cN1/2 diseases, who became or remained ycN0 at the end of neoadjuvant treatment and underwent surgical treatment. 

We excluded from this study four cN0–ycN0 patients and nine cN+-ycN0 patients that underwent immediate ALND following a non-identification of a SLN during axillary procedure. The overall SLN identification rate was 96.8% (98.2% in cN0-ycN0 patients and 95.2% in cN+-ycN0 patients).

Regarding the remaining 399 cases that underwent SLNB, in 117 (29.4%) the main indication for NACT was to reduce the tumor diameter and achieve a conservative breast treatment instead of a mastectomy; 104 (26.0%) had the presence of clinically involved lymph nodes (cN+) and 76 cases (19.0%) both concomitant situations.

Furthermore, in 102 cases (25.6%) patients younger than 50 years with unfavorable risk factors received NACT mainly to ensure a prompt systemic treatment, independently of T/N status.

Among patients that underwent NACT, 219 patients that were cN0 at the time of diagnosis remained ycN0, while 180 cN1/2 patients benefit of chemotherapy and down staged to ycN0 status (patients characteristics prior to neoadjuvant chemotherapy, according to axillary clinical status before systemic treatment are summarized in Table 1).

Clinical restaging after NACT: Neoadjuvant regimes and clinical response are summarized in Table 2. Concerning clinical response, we observed an overall complete clinical response in 144 patients (36.1%), a partial response in 228 patients (57.1%) no response in 12 patients (3%) and a progression to T stage, with breast skin or pectoralis major fascia involvement, in 15 patients (3.8%).

Among women treated with hormone-based NACT, we observed four cases of clinical complete response to treatment (16.7%), 15 cases of partial response (62.4%) and five cases of no response (20.9).

Breast surgery: 246 patients received conservative OPS and 153 women were given mastectomy (134 patients (87.6%) were treated with conservative mastectomy followed by implant/autologous reconstruction).

Among elderly and fragile patients treated with hormone-based NACT, four patients (16.7%) with initial skin involvement did not experience any response to NACT and were consequently treated with a simple mastectomy (Table 3).

Axillary treatment: During SLNB surgical procedure a mean number of 2.7 lymph nodes (3, 1–7) were identified and removed.

Overall, SLNB was negative in 235 cases (58.9%). Among 219 cN0-ycN0 patients, 149 (68%) were confirmed ypN0(sn). Of the 180 cN1/2-ycN0 cases, 86 (47.8%) confirmed an effective downstaging to an ypN0 (sn) status, 76 patients (42.2%) remained macro-metastatic, 10 patients (5.6%) decreased to a micro-metastatic involvement and eight patients (4.4%) patient revealed a residual ITC positivity. 

Among patients given ALND, a mean number of 12.8 lymph nodes (11.5, 6–30) were removed. Axillary pathological staging is summarized in Table 4.

Pathological Characteristics: Pathological characteristics are shown in Table 5. Complete breast remission (ypT0) occurred in 132 (33.1%) women. Tumors were luminal A-like, luminal B-like, HER2 positive and triple negative in 75 (18.8%), 121 (30.4%), 10 (2.5%) and 33 (8.2%) cases, respectively. Tumor subtype was not assessable in 160 (40.1%) patients with complete pathological remission in the breast or very limited residual disease.

Oncological outcomes: After a median of 35.6 months (2–55), axillary failure (AF) occurred in 2 cN0 patients with a negative SLN and in 4 cN1/2 patients with a negative SLNB.

In our experience, AF occurred in patients that were diagnosed in unfavorable conditions, such as multifocal (two patients; 33%), cT3 (two patients; 33%) and T4b (one patient; 16.5%) tumors. Moreover, we observed an AF in 1 patient with ITC SLN positivity, who refused both ALND and adjuvant axillary radiotherapy. 

Furthermore, in 3/6 patients, AF was diagnosed concurrently to a distant relapse (50%).

Overall survival: During the entire follow-up, we reported the death of 15 (3.8%) women: two in the SN-negative group (OS 97.4%) and 13 in the SN-positive group (OS 82.7%)—*p* < 0.0001. Death was attributed to breast cancer in 92.5% of cases. Three-year OS was 94.3% overall, 95.5% in those initially cN0 and 93% in those initially cN1/N2.Distant disease free survival: Overall 36 (9%) patients developed distant metastases (DDFS 83.8%). According to SN-status we report six patients with distant metastasis in SN-negative group (DDFS 95.7%) and 30 patients in the SN-positive group (DDFS 67.9%)—*p* < 0.0001. Three-year DDFS was 92.2% in those initially cN0 and 84.8% in those initially cN1/2.Regional Disease Free survival: Overall, 24 patients developed a regional recurrence (RFS 89.4%): eight (2%) women had ipsilateral breast cancer recurrence, two (0.5%) had contralateral breast cancer, and 10 (2.5%) patients developed axillary recurrence. In four (1%) patients we diagnosed a synchronous recurrence in breast and axilla. RDFS was 96.5 % in patients with negative SLNB and 91.3% in those with positive SLNB (*p* = 0.007). Three-year RDFS was 94.2% in those initially cN0 and 87.9% in those initially cN1/2.

OS, DDFS and RDFS were significantly related to SLNB positivity, even overall, and according to axillary staging before NACT (cumulative incidence of regional relapses, as well as OS and DDFS curves, are shown in Figure 2).

At uni- and multivariate analysis (Table 5), positive SLN was confirmed as an independent prognostic factor for DDFS, as well as triple negative immunophenotype and T pathological complete response, even in the cN0-ycN0 and in the c1/2-ycN0 group. 

## 6. Discussion

In an effort to minimize the clinical impact of breast cancer, several improvements have been made with regards to breast surgical treatment following NACT [14,15,16].

Conversely, the axillary approach remains a controversial field: SLNB is considered the gold standard for axillary staging in early breast cancer patients with clinically negative lymph nodes, confining ALND to a very limited group of patients.

The purpose of this de-escalation in surgery is to reduce axillary morbidity (seroma formation, loss of arm sensitivity, shoulder dysfunction and lymphedema) by restricting or avoiding axillary dissection without proven oncological advantages.

However, to be reliable SLNB should always be over 90% in identification rate (IR) and below 10% in false negative rate (FNR), conditions that could be easily met in early breast cancer treatment, whereas after NACT initial experiences reported questionable results [17,18].

Despite these initial observations, a progressive set-up of the axillary workout before and after NACT led to metanalysis of retrospective studies that seem to validate SLNB after NACT, reporting acceptable IR and FNR, comparable to those reported for the early breast cancer setting [19,20,21].

Moreover, recent evidence also seemed to validate SLNB in ycN0 patients, for whom positivity would also play an important role as a significant prognostic factor [13]. 

We analyzed records regarding 399 consecutively treated patients. We achieved, even with a single agent technique, an acceptable IR for cN0-ycN0 and for cN1/2-ycN0 patients (98.2% and 95.2%, respectively).

These data are in line with previously published results in theoretically more favorable conditions, such as its execution in early breast cancer setting, and obtained with the use of double tracer (radiotracer + blue dye), thus strengthening the observation of Fringuelli et al. [22] that NACT does not influence axillary lymphatic drainage and consequently axillary mapping success, furthermore clearly confirming that identification rate of SLNs actually improves with the surgical experience of the operating team, especially for single tracer technique, as reported by Zhang in the neoadjuvant setting [23].

Regarding prognostic power of SLNB after NACT, the European Institute of Oncology recently published a paper in which Galimberti et al. analyzed 396 cT1-4, cN0/1/2 patients who became or remained cN0 after neoadjuvant treatment and underwent SLNB.

Their data confirmed SLN status as a significant prognostic factor in cN0-ycN0 patients, a finding that seems to be consistent with the known prognostic significance of axillary involvement in the early breast cancer setting. However, at multivariate analysis, SLNB lost its prognostic power in the cN1/2-ycN0 group, suggesting that an axillary involvement before NACT could potentially jeopardize a reliable mini-invasive radiation.

Our data (although a result of limited and preliminary observations) confirm Galimberti’s conclusions: in 219 cN0-ycN0 patients SLNB was safely performed. In this subgroup, we observed two cases of axillary failure, and three-year OS, DDFS and RDFS were 95.5%, 92.2% and 94.2%, respectively. Moreover, multivariate analysis showed that strong prognostic factors such as triple negative immunophenotype, persistence of extended involvement of the axilla (ypN2/3) and positivity of SLNB maintain a statistically comparable prognostic role.

In this setting, our observations further reinforce not only the feasibility but also the low risk of false negative rates of SLNB, confirming its predictive role even after NACT.

Among 180 cN1/2–ycN0 patients, a mini-invasive axillary staging by means of SLNB should be taken into account both for the high identification rate and also for the observed axillary complete pathological response rate (in our experience 47.8%).

In such a setting, although there is a higher rate or axillary failure (4.6% versus 1.3% in cN0-ycN0 patients), we registered three-year OS, DDFS and RDFS rates that were comparable to those registered in cN0 patients (93%, 84.8% and 87.9%, respectively).

We also confirmed SLNB’s prognostic power at multivariate analysis that, even in a more complex subgroup of patients, resulted in statistical comparability to other strong prognostic factors such as triple negative immunophenotype, and complete pathological response on the breast (ypT0).

This observation differs from Galimberti’s conclusions for cN1/2–ycN0 patients, a phenomenon that can be related to our shorter follow up timing, suggesting that in the first three years axillary response could reflect a systemic control of disease, but also to our high rate of patients diagnosed with cN1 axillary status before NACT (144/180 (80%) of cases included in cN1/2 group were cN1).

This particular subset distribution could have influenced uni- and multivariate results, suggesting that SLNB remains a reliable prognostic tool in patients with a lower grade of initial axillary involvement, whereas in patients with a major axillary burden prior to neoadjuvant treatment, the disruption of the lymphatic architecture (caused both by both perinodal infiltration and chemo-therapic agents) could compromise the reliability of SLN in predicting axillary status and therefore compromise its predictive prognostic power.

## 7. Conclusions

Our results, although from a single institution and being a retrospective experience with a limited follow-up timing, strengthen the possibility of safely performing SLNB after NACT in cN0-ycN0 patients, and reinforce the need for further refinements in the mini-invasive axillary approach for cN1-2 patients who become node-negative after NACT.

## Figures and Tables

**Figure 1 jpm-11-00172-f001:**
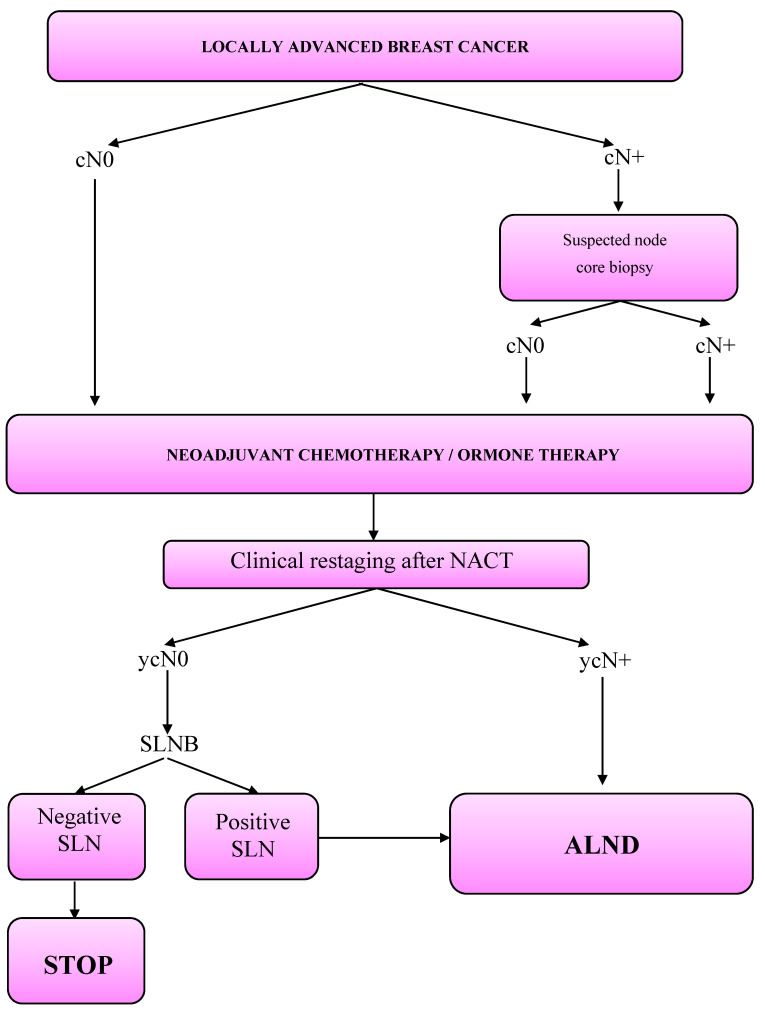
Axillary workout in the neoadjuvant setting.

**Figure 2 jpm-11-00172-f002:**
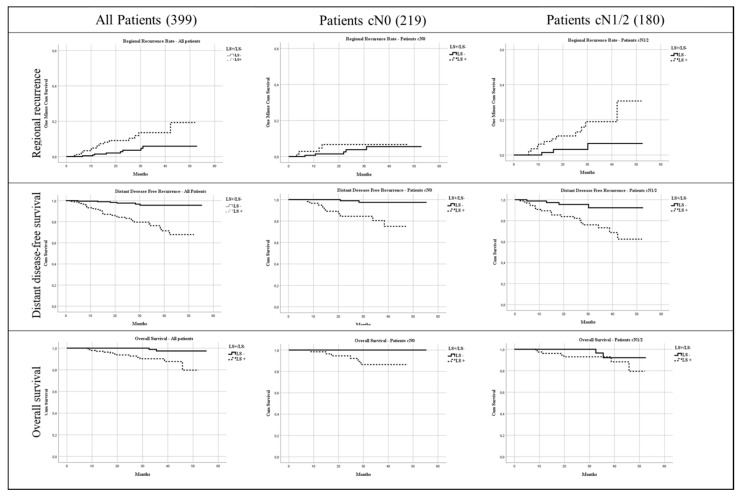
Cumulative Overall Survival, Regional Recurrence and Distant Disease Free Survival.

**Table 1 jpm-11-00172-t001:** Clinical characteristics of 399 patients according to cN status prior to neoadjuvant treatment.

	cN0	cN1/2
All	219 (54.8%)	180 (45.2%)
Age (years)		
<35	21 (9.6%)	12 (6.7%)
35–49	105 (47.9%)	89 (49.4%)
50–69	80 (36.5%)	70 (38.9%)
>70	13 (5.9%)	9 (5%)
Breast Related Cancer Antigens (BRCA) mutations	29 (13.2%)	16 (47.2%)
Menopausal status	103 (47%)	85 (47.2%)
Grading		
G1	3 (1.4%)	2 (1.1%)
G2	78 (35.6%)	66 (36.7%)
G3	118 (53.9%)	96 (53.3%)
Unknown	20 (9.1%)	16 (8.9%)
Tumor subtype		
Luminal A	8 (3.7%)	3 (1.7%)
Luminal B	152 (69.4%)	133 (73.9%)
HER 2 positive	17 (7.8%)	16 (8.9%)
Triple negative	42 (19.2%)	28 (15.6%)
Clinical T		
cT1	28 (12.8%)	27 (15%)
cT2	146 (66.7%)	105 (58.3%)
cT3	29 (13.2%)	33 (18.3%)
cT4	16 (7.3%)	15 (8.3%)
Multifocality/multicentricity	91 (41.6%)	92 (51.1%)

**Table 2 jpm-11-00172-t002:** Schemes of delivered neoadjuvant treatments according to axillary clinical stage at diagnosis and related clinical response.

	cN0	cN1/2
All	219 (54.8%)	180 (45.2%)
Neoadjuvant treatment		
Hormone Therapy	23 (10.5%)	1 (0.6%)
Chemotherapy	196 (89.5%)	179 (99.4%)
Neoadjuvant chemotherapy		
Anthracycline and/or Taxane	5 (2.6%)	4 (2.2%)
Anthracycline + Taxane	159 (81%)	141 (78.8%)
Other	32 (16.4%)	34 (19%)
Herceptin containing regimen	64 (29.2%)	59 (32.8%)
Clinical response		
Complete response	77 (35.2%)	67 (37.2%)
Partial response	125 (57%)	103 (57.3%)
No response	8 (3.7%)	4 (2.2%)
Progression	9 (4.1%)	6 (3.3%)

**Table 3 jpm-11-00172-t003:** Surgical treatment and adjuvant radiotherapy.

	cN0	cN1/2
All	219 (54.8%)	180 (45.2%)
Surgery		
Conservative surgery	142 (64.8%)	104 (57.8%)
Conservative mastectomy	68 (31.1%)	66 (36.7%)
Simple mastectomy	9 (4.1%)	10 (5.6%)
RT after conservative surgery		
No treatment *	6 (4.2%)	3 (2.9%)
Radiotherapy	136 (95.8%)	101 (97.1%)
RT after mastectomy		
No treatment	48 (62.3%)	16 (21.1%)
Radiotherapy	29 (37.7%)	60 (78.9%)

* Patient’s refusal or early progression of systemic disease.

**Table 4 jpm-11-00172-t004:** Pathological characteristics according to cN status prior to neoadjuvant treatment.

	cN0	cN1/2
All	219 (54.8%)	180 (45.2%)
ypT		
ypT0	66 (30.2%)	66 (36.7%)
ypTmic	20 (9.1%)	24 (13.3%)
ypT1	92 (42%)	64 (35.5%)
ypT2	36 (16.4%)	21 (11.7%)
ypT3	3 (1.4%)	3 (1.7%)
ypT4	2 (0.9%)	2 (1.1%)
Multifocality/multicentricity	55 (25.1%)	44 (24.4%)
ypN		
ypN0	149 (68%)	86 (47.8%)
ypNi+ *	13 (5.9%)	11 (6.1%)
ypNmic **	18 (8.2%)	10 (5.6%)
ypN1	34 (15.5%)	55 (30.6%)
ypN2	5 (2.3%)	17 (9.4%)
ypN3	0 (0%)	1 (0.6%)
ER		
Positive	112 (51.1%)	80 (44.5%)
Negative	30 (13.7%)	17 (9.4%)
Not evaluable ***	77 (35.2%)	83 (46.1%)
PR		
Positive	83 (37.9%)	50 (27.8%)
Negative	59 (26.9%)	47 (26.1%)
Not evaluable ***	77 (35.2%)	83 (46.1%)
Ki-67		
<24%	94 (42.9%)	64 (35.6%)
≥25%	48 (21.9%)	33 (18.3%)
Not evaluable ***	77 (35.2%)	83 (46.1%)
Tumor subtype		
Luminal A	46 (21%)	29 (16.1%)
Luminal B	70 (32%)	51 (28.4%)
HER2	4 (1.8%)	6 (3.3%)
Triple negative	22 (10%)	11 (6.1%)
Not evaluable ***	77 (35.2%)	83 (46.1%)

* evidence of isolated cancer cells in the lymph node. ** evidence of microscopic residual of tumor (<0.2 mm) in the lymph node. *** ypN0, ypN1mic and ypNi+.

**Table 5 jpm-11-00172-t005:** Univariate and multivariate analysis for Distant Disease Free Survival.

	All Patients		cN0		cN1/2	
	Univariate Analysis	Multivariate Analysis	Univariate Analysis	Multivariate Analysis	Univariate Analysis	Multivariate Analysis
**Clinical Characteristics**						
Menopausal status	0.804	/	0.861	/	0.139	/
BRCA1/2 mutation	0.430	/	0.460	/	0.996	/
Multifocality ad the diagnosis	0.288	/	0.430	/	0.811	/
Luminal HER2 +	0.524	/	0.720	/	0.504	/
Triple Negative	1.231 (0.002)	1.879 (0.0001)	1.394 (0.023)	2.606 (0.0001)	1.668 (0.027)	1.888 (0.002)
**Pathological Characteristics**						
ypT2, ypT3, ypT4	0.873 (0.014)	0.767	0.071	/	0.925 (0.040)	0.417
LS + (ypN+(sn))	2.048 (0.0001)	1.977 (0.0001)	2.502 (0.001)	2.807 (0.001)	1.540 (0.005)	1.213 (0.045)
ypN2, ypN3	1.946 (0.0001)	1.370 (0.003)	2.759 (0.0001)	2.157 (0.004)	1.237 (0.0027)	0.121
pCR on T	−1.815 (0.003)	0.331	−3.704 (0.142)	/	−1.432 (0.020)	0.457

## Data Availability

The data presented in this study are available on request from the corresponding author.

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
