# Peer review of "Sentinel Node Biopsy after Neoadjuvant Chemotherapy for Breast Cancer: Preliminary Experience with Clinically Node Negative Patients after Systemic Treatment"

_jpm, 2021, doi:10.3390/jpm11030172_

Round 1
Reviewer 1 Report
A better explanation of the purpose of the study, in the introduction, is needed, in my opinion. First of all, you should clarify the role of SLNB in the subsetting you have identified and its prognostic power either in negative or in positive node metastasis (you talk about this topic at the end of the article but I think that it could be emphatized the most); furthermore, it could be interesting if you could analyze the survival tools regarding the entity of node metastasis: did you give ALND to all positive SLNB (either micro- or macrometastasis)? And, in this case, did NACT affect SLNB?
Author Response
Reviewer 1
- A better explanation of the purpose of the study, in the introduction, is needed, in my opinion. First of all, you should clarify the role of SLNB in the subsetting you have identified and its prognostic power either in negative or in positive node metastasis (you talk about this topic at the end of the article but I think that it could be emphatized the most)
Thank you for your brilliant suggestion. We clarified the effective role of SLNB after NACT in the introduction section. We hope you will find it more clear and oriented that it was before.
- It could be interesting if you could analyze the survival tools regarding the entity of node metastasis: did you give ALND to all positive SLNB (either micro- or macrometastasis)? And, in this case, did NACT affect SLNB?
Thank you for your comment.
We better clarified our axillary workout in Materials and methods (on regards of axillary assessment): “For patients with ypN+ (micro or micrometastatic) disease, axillary dissection was directly performed. I and II levels lymph nodes were always removed, while III level lymph nodes were removed only in case of in case of intraoperative detection of clinically suspicious nodes at lower levels. Patients with isolated tumor cells positivity at SLNB were treated as ypN0 and did not receive ALND.
Regarding the effect of NACT on SLN detection, we achieved an acceptable IR for cN0-ycN0 and also for cN1/2-ycN0 patients (98.2 and 95.2% respectively) that can be comparable with non neoadjuvated patients.
We can therefore conclude that in our experience, NACT did not impact on SLN detection.
Regarding the pathologic effect observed on cN+ patients, we observed a pathological conversion rate to ypN0 in 47.8% of patients, 5.6% decreased to a micrometastastic involvement and 4.4% patient revealed a residual ITC positivity (changes were made in the text, among RESULTS, the “axillary treatment” section).
Reviewer 2 Report
Interesting subject but many papers have ben written on this
The modern approach of clipping the affected node and removal following chemo is used in many units
May need to add some patients using this technique and give better value to the paper
Author Response
Reviewer 2
- Interesting subject but many papers have been written on this. The modern approach of clipping the affected node and removal following chemo is used in many units. May need to add some patients using this technique and give better value to the paper.
Thank you for your comments. It is true that many articles have been written about SLNB after NACT, but most of these experiences are oriented to validate this procedure in terms of reliability (basing on identification rates) and as a predictive tool of axillary status.
In our opinion, there are some aspects that are still unclear, where literature still presents limited and conflicting results, such as the effective value of SLN status as a prognostic factor after NACT, a topic treated only by Galimberti (3).
It is mainly with this purpose that we wanted to present our preliminary results, because we are strongly convinced that with the right multidisciplinary workout, before, during and after neoadjuvant treatments, SLNB could be as effective and predictive as when performed in non neoadjuvated patients.
Regarding the use of markers positioned in metastatic lymph nodes prior to NACT, we introduced this procedure in clinical routine in June 2019.
We observed 18% of cases in which there was not an intraoperative coincidence between clipped lymph node and SLN. As a result of this limited experience, to avoid confusing results we decided to exclude clipped patients from our analysis, focusing mainly on patients that received SLNB as the only axillary procedure.